# Combined Nivolumab and Ipilimumab in Octogenarian and Nonagenarian Melanoma Patients

**DOI:** 10.3390/cancers15174330

**Published:** 2023-08-30

**Authors:** Constance Reichert, Capucine Baldini, Sarah Mezghani, Eve Maubec, Christine Longvert, Laurent Mortier, Gaëlle Quereux, Arnaud Jannic, Laurent Machet, Julie de Quatrebarbes, Charlée Nardin, Nathalie Beneton, Mona Amini Adle, Elisa Funck-Brentano, Vincent Descamps, Lorry Hachon, Nausicaa Malissen, Barouyr Baroudjian, Florence Brunet-Possenti

**Affiliations:** 1Department of Dermatology, Hôpital Bichat AP-HP, Université Paris Cité, 75018 Paris, France; constance.reichert@aphp.fr (C.R.); vincent.descamps@aphp.fr (V.D.); 2Drug Development Department, Institut Gustave Roussy, CNRS-UMS 3655 and INSERM US23, 94805 Villejuif, France; capucine.baldini@gustaveroussy.fr; 3Department of Imaging, Institut Curie, PSL Research University, 75005 Paris, France; sarahmezghani@gmail.com; 4Department of Dermatology, Hôpital Avicenne AP-HP, Université Sorbonne Paris Nord—Campus de Bobigny, 93000 Bobigny, France; eve.maubec@aphp.fr; 5Department of Dermatology, EA4340-BECCOH, Hôpital Ambroise Paré APHP, Université Paris-Saclay, 92100 Boulogne-Billancourt, France; christine.longvert@aphp.fr (C.L.); elisa.funck-brentano@aphp.fr (E.F.-B.); 6Department of Dermatology, Claude Huriez Hospital, Lille University, Inserm U1189, 59000 Lille, France; laurent.mortier@chu-lille.fr; 7Department of Dermatology, Centre Hospitalier Universitaire de Nantes, CIC 1413, INSERM, Immunology and New Concepts in ImmunoTherapy, INCIT, UMR 1302, Nantes University, 44000 Nantes, France; gaelle.quereux@chu-nantes.fr; 8Dermatology Department, Hôpital Henri Mondor AP-HP, 94000 Créteil, France; arnaud.jannic@aphp.fr; 9Department of Dermatology, Tours University Hospital, 37000 Tours, France; machet@univ-tours.fr; 10Department of Dermatology, Centre Hospitalier Annecy-Genevois, 74370 Annecy, France; jdequatrebarbes@ch-annecygenevois.fr; 11Université de Franche-Comté, CHU Besançon, EFS, INSERM, UMR RIGHT, 25000 Besançon, France; cnardin@chu-besancon.fr; 12Department of Dermatology, Centre Hospitalier du Mans, 72037 Le Mans, France; nbeneton@ch-lemans.fr; 13Oncodermatology Department Centre Léon Bérard, 69008 Lyon, France; mona.amini-adle@lyon.unicancer.fr; 14Department of Pharmacy, Hôpital Bichat, AP-HP, 75018 Paris, France; lorry.hachon@outlook.fr; 15Dermatology and Skin Cancer Department, APHM, CRCM Inserm U1068, CNRS U7258, CHU Timone, Aix Marseille University, 13007 Marseille, France; nausicaa.malissen@ap-hm.fr; 16Department of Dermato-Oncology, Hôpital Saint-Louis AP-HP, Inserm U976, Université Paris Cité, 75010 Paris, France; barouyr.baroudjian@aphp.fr

**Keywords:** melanoma, elderly, octogenarian, nonagenarian, immune checkpoint inhibitors, anti-PD-1, anti-CTLA-4

## Abstract

**Simple Summary:**

Elderly cancer patients over the age of 80 years represent a growing population; these are notably melanoma patients since 25% of cases are diagnosed after 75 years of age. Establishing the best therapeutic strategies for older patients with melanoma is a challenge since this subpopulation has poor disease-specific outcomes, partly due to age-related variations in therapeutic management. Regarding mono-immunotherapy, either with anti-CTLA-4 or anti-PD-1, several studies have shown similar tolerability outcomes in older and younger patients. However, there are limited data in very elderly patients, especially those treated with immunotherapy combinations. The aim of this multicenter retrospective study is to analyze the prescribing patterns and the safety profile of nivolumab combined with ipilimumab in a cohort of octogenarian and nonagenarian melanoma patients in a real-life setting. The results indicate that the prescribing patterns are very heterogeneous in patients aged over 80 years old, highlighting the lack of clear therapeutic guidelines in the elderly population. In this cohort, the toxicity data did not show a high frequency of severe immune-related adverse events.

**Abstract:**

Data regarding elderly melanoma patients treated with anti-PD-1 or anti-CTLA-4 antibodies are in favor of tolerability outcomes that are similar to those of younger counterparts. However, there are very few studies focusing on elderly patients receiving nivolumab combined with ipilimumab (NIVO + IPI). Here, we ask what are the current prescribing patterns of NIVO + IPI in the very elderly population and analyze the tolerance profile. This French multicenter retrospective study was conducted on 60 melanoma patients aged 80 years and older treated with NIVO + IPI between January 2011 and June 2022. The mean age at first NIVO + IPI administration was 83.7 years (range: 79.3–93.3 years). Fifty-five patients (92%) were in good general condition and lived at home. Two dosing regimens were used: NIVO 1 mg/kg + IPI 3 mg/kg Q3W (NIVO1 + IPI3) in 27 patients (45%) and NIVO 3 mg/kg + IPI 1 mg/kg Q3W (NIVO3 + IPI1) in 33 patients (55%). NIVO + IPI was a first-line treatment in 39 patients (65%). The global prevalence of immune-related adverse events was 63% (38/60), with 27% (16/60) being of grade 3 or higher. Grade ≥ 3 adverse events were less frequent in patients treated with NIVO3 + IPI1 compared with those treated with NIVO1 + IPI3 (12% versus 44%, *p* = 0.04). In conclusion, the prescribing patterns of NIVO + IPI in very elderly patients are heterogeneous in terms of the dosing regimen and line of treatment. The safety profile of NIVO + IPI is reassuring; whether or not the low-dose regimen NIVO3 + IPI1 should be preferred over NIVO1 + IPI3 in patients aged 80 years or older remains an open question.

## 1. Introduction

Nivolumab combined with ipilimumab (NIVO + IPI) is a highly effective treatment in advanced melanoma. This combination is associated with durable and improved outcomes compared with those of nivolumab alone in the descriptive analysis of the Checkmate 067 trial, with a median overall survival of 72.1 months with NIVO + IPI versus that of 36.9 months with NIVO alone [1]. Nevertheless, the incidence of immune-related adverse events (irAE) was higher in the NIVO + IPI group with 95.5% of patients who experienced any-grade irAE versus that of 82.1% of patients in the NIVO group [2]. Nowadays, NIVO + IPI often constitutes the preferred first-line treatment for patients in good general condition [3], particularly for those with metastatic mucosal melanoma [4] or brain metastases [5] because of their dark prognosis.

With our aging population, improving the management of cancer in older persons is a public health challenge [6]. This is especially true for melanoma since 25% of cases are diagnosed after 75 years [7] and also because this subpopulation displays poor disease-specific outcomes, partly due to age-related variations in clinical management [8]. Furthermore, elderly patients with metastatic melanoma are often autonomous and in good clinical condition at the time of diagnosis, thus being willing to receive optimal treatment.

Regarding mono-immunotherapy, either anti-CTLA-4 or anti-PD-1, several studies have shown similar efficacy and tolerability outcomes in older and younger populations, regardless of tumor type [9,10] as well as in series focusing on melanoma [11,12]. However, there are limited data on patients aged 80 years or older, since the threshold used for defining older adults is often 65 years [13], which is not representative of the very elderly subgroup. Moreover, elderly patients are underrepresented in clinical trials [14] and those included in trials are usually very fit persons.

Assessing the benefit:risk ratio of cancer immunotherapies for octogenarian and nonagenarian patients is crucial because of several concerns related to toxicity in this specific population [15]. First, advanced age is associated with immunosenescence, which is responsible for increased serum levels of autoantibodies [16]; in this context the introduction of immune checkpoint inhibitors is likely to reveal subclinical autoimmune diseases [17]. Another concern is the potential severe impact of immune-related adverse events (irAE) on elderly patients who are often polymedicated, with multiple comorbidities including renal failure and cognitive dysfunction. For example, colitis, even in a mild case, can rapidly lead to severe dehydration and/or can be complicated by hemorrhage in patients treated with blood thinners [18]. The risk of severe complications is even higher in socially isolated patients. Additionally, the treatments used for irAE can be harmful for older patients; glucocorticoids in particular can worsen underlying comorbidities such as diabetes, HTA or cognitive impairment which are frequent in this population [19]. Finally, hospitalization may have a deleterious effect, resulting in a loss of autonomy.

This study aims to describe the prescribing patterns and the safety profile of NIVO + IPI in a cohort of octogenarian and nonagenarian melanoma patients in a real-life setting.

## 2. Materials and Methods

### 2.1. Study Participants

This was a retrospective French study that included melanoma patients aged 80 years or older treated with NIVO + IPI in 13 cancer centers between January 2011 and June 2022.

### 2.2. Data Collection

Data were collected through the French network Groupe de Cancérologie Cutanée, using electronic medical records. Baseline characteristics included age, gender, Eastern Cooperative Oncology Group performance status (ECOG PS), LDH level, the presence of hypertension, diabetes, overweight status, dementia or a chronic inflammatory condition. Tumor characteristics included the melanoma subtype, BRAF mutation status and stage according to American Joint Committee on Cancer, 8th edition criteria. Treatment history included the number and type of previous lines of systemic curative treatment. Treatment modalities included the dosing regimen and the number of infusions. Tolerability data included the type of irAE, time of onset, grade according to Common Terminology Criteria for Adverse Events v5.0, number of irAE and their outcome. Clinical and/or radiological outcomes were those reported at the first visit following treatment discontinuation.

### 2.3. Statistical Analysis

Quantitative variables were expressed as median [range] and mean (standard deviation), and compared using a Student’s *t* test. Qualitative variables were expressed as numbers and percentages and compared using either Fisher’s exact test or the Chi-2 test depending on the size of the group. The statistical analyses were performed with R studio (version 1.2.1335, Vienna, Austria). All tests were two-tailed, and statistical significance was determined by a *p* value of <0.05.

## 3. Results

### 3.1. Patients’ Characteristics

In total, 60 patients (37 males and 23 females) were included in the study; their characteristics are reported in Table 1. The mean age at first NIVO + IPI infusion was 83.7 years (SD 3.2) (range: 79.3–93.3). Most patients had an ECOG PS of ≤ 1 (47/60, 78%), were both independent and were living and home (55/60, 92%). Fifty-six patients (93%) had at least one chronic disease which was either hypertension, overweight or diabetes and three presented mild dementia. Twenty patients (33%) were polymedicated (defined as five drugs or more being prescribed simultaneously). The primary melanoma was cutaneous (37/60, 62%), mucosal (12/60, 20%), acral (3/60), uveal (1/60) or unknown (7/60). Patients were treated for stage IV (50/60, 83%) or unresectable stage II or III (10/60) melanoma. Five patients had a BRAF mutation. Among the 50 stage IV patients, 22 had cerebral metastases (37%). Baseline LDH levels were ≥2 ULN for 9 patients (15%).

### 3.2. Prescribing Patterns of NIVO + IPI

Two dosing regimens were used: the standard regimen that is NIVO 1 mg/kg + IPI 3 mg/kg Q3W (NIVO1 + IPI3) in 27 patients (45%) and the low-dose IPI regimen that is NIVO 3 mg/kg + IPI 1 mg/kg Q3W (NIVO3 + IPI1) in 33 patients (55%). Baseline characteristics were similar in the two groups, except for age; patients treated with NIVO1 + IPI3 were slightly younger than those in the NIVO3 + IPI1 group, with a mean age of 82.4 versus 84.7 (*p* = 0.002). All four nonagenarian patients received NIVO3 + IPI1. NIVO + IPI was administered as a first-line treatment in 39 patients (65%), including the 12 patients with mucosal melanoma.

Twenty-one patients (35%) had previously received at least one line of curative-intent treatment that always included an anti-PD-1 therapy but no previous IPI therapy (Table 1). Among the 27 patients treated with NIVO1 + IPI3, 25 received it as a first-line therapy (25/27, 93%) whereas NIVO3 + IPI1 was a first line in 42% of patients treated with this regimen (14/33).

Among the 13 participating centers, 4 systematically administered NIVO + IPI both as a first-line treatment and with the standard dosing regimen, while various practices were reported by the other centers.

### 3.3. Immune-Related Adverse Events

The global prevalence of irAE was 63% (38/60) with 27% being of grade 3 or higher (Table 2). The mean time from NIVO + IPI initiation to irAE onset was 36 days (range: 3–109); 51% of irAE occurred during the first month of treatment. The most frequent toxicities were skin rashes (13/60), colitis (12/60), endocrinopathies (11/60) and hepatitis (8/60) (Table 3). Compared to NIVO3 + IPI1, the irAE in the NIVO1 + IPI3 group had an earlier onset (23 versus 49 days, *p* = 0.003) and were more frequently severe (44% versus 12% of grade ≥ 3, *p* = 0.04). Seventeen patients stopped NIVO + IPI due to toxicity, with 48% in the NIVO1 + IPI3 group versus 12% in the NIVO3 + IPI1 group (*p* = 0.003).

Focusing on the 39 patients treated in the first-line setting (Table 4), similar data were observed with more frequent grade ≥ 3 irAE in the NIVO1 + IPI3 group (48% versus 7%). Two male patients without any cardiac history, aged 82 and 84 years, respectively, died of immune-related myocarditis following NIVO1 + IPI3 initiation as a first-line therapy. For the first patient, myocarditis occurred 19 days after the first infusion, with a fatal outcome at day 25; the second patient developed symptoms after the third infusion and died two weeks later. One case of Horton’s disease causing irreversible blindness occurred in a female patient after the fourth infusion of NIVO1 + IPI3.

Nine of the patients previously treated with anti-PD-1 had a history of mild irAE (grade 1–2). Among them, four experienced irAE recurrences that were all mild. Two patients treated with NIVO3 + IPI1 had a history of chronic inflammatory disease, one with cutaneous psoriasis that worsened along with the onset of arthritis, successfully treated with methotrexate and oral prednisone; the other patient had rheumatoid arthritis treated with oral steroids that remained stable. Both patients were able to receive the four cycles of NIVO + IPI.

Ten patients received concomitant radiotherapy (five of which received it for cerebral metastases, and the other five received it for skin and/or lymph node metastases) with good tolerance, except for one case of severe radiation dermatitis that resolved within 2 months along with the regression of skin metastases.

### 3.4. Patients’ Outcomes at First Evaluation

Assessing NIVO + IPI efficacy was not an objective of this study because of the marked heterogeneity in prescribing patterns and the small size of the cohort. Moreover, the various modalities in terms of time and methods used for assessing tumor response make the comparison between the two dosing groups irrelevant. Herein, we report the observational data related to the first evaluation performed after NIVO + IPI discontinuation in the 39 patients with first-line treatment (Table 5): no complete response, 15 partial responses (38.5%), 3 cases of disease stability (8%), 15 cases of progression (38.5%), 4 melanoma deaths (10%) and 2 irAE-related deaths (5%).

Among the 60 patients included in the study, nine deaths related to disease progression occurred during the first 30 days of treatment in patients previously treated with multiple lines of treatment (four out of nine) and/or with an ECOG PS 2–3 (five out of nine). Of note, among the 13 patients with an ECOG PS of ≥ 2 who initiated NIVO + IPI 10 were either dead or had progressive disease at the time of the first evaluation.

## 4. Discussion

To our knowledge, this is the first study focusing on the modalities of use of NIVO + IPI in octogenarian and nonagenarian patients. The majority of patients (35/60, 58%) received the combination immunotherapy with low-dose ipilimumab and/or as a salvage treatment. These results indicate that the prescribing patterns of NIVO + IPI are very heterogeneous in melanoma patients aged over 80 years, highlighting the lack of clear therapeutic guidelines for this subpopulation. These various patterns might reflect physicians’ concerns regarding the risk of irAE occurrence, due to limited data on the use of NIVO + IPI in the very elderly populations [20].

In our study, the overall irAE rate was 63% and only one quarter of patients experienced grade ≥ 3 irAE, which is much less than the occurrence of that in the pivotal trial of NIVO + IPI that reported the occurrence of grade 3–4 irAE in 55% of patients [2]. Similarly, a recent retrospective study conducted on melanoma patients aged over 75 years reported 42% of irAE being of grade 3 or higher [21]. Herein, the low rates of irAE are partly explained by the study population that included 35% of patients previously treated with anti-PD-1 with good tolerance. Additionally, there were nine patients in poor general condition and/or with a history of multiple lines who died of rapid progression within the first month; hence, they had less time to develop irAE. Furthermore, 55% of patients received the low-dose IPI regimen and as expected the prevalence of severe irAE was significantly lower in patients treated with NIVO3 + IPI1 compared to that in patients treated with NIVO1 + IPI3 (12% versus 44%). These results are in line with those of other studies comparing the two regimens, such as the CheckMate 511 trial that reported 86% of all-grade toxicity but only 33% of grade 3–4 with NIVO3 + IPI1 [22].

Even though the available studies related to the use of combination anti-PD-1 + low-dose ipilimumab support the robust antitumor activity of this regimen [23], these trials were underpowered in terms of efficacy [24]. Hence, NIVO3 + IPI1 cannot be recommended over NIVO1 + IPI3 in the metastatic setting [25]. In very old patients, the benefit:risk ratio differs slightly from that of younger counterparts, and it may seem more appropriate to systematically prescribe the low-dose regimen. However, very advanced age is associated with immunosenescence [17] and some studies have suggested that anti-PD-1 monotherapy may be less effective in patients over 75 years [26]. While there could be no or little difference in efficacy between the low- and full-dose regimen in young patients, there may be significant difference in elderly patients. In the absence of data related to NIVO + IPI in this population, we consider that the choice of regimen should be guided by the evaluation of several geriatric domains, following the recommendations of the International Society of Geriatric Oncology (SIOG) [27]: demographic data and social support, functional status, cognition, mental health, nutritional status, fatigue, comorbidities, polypharmacy and other geriatric syndromes, as well as an evaluation of quality of life. The line of treatment should also be taken into account. For example, if a patient is considered fit enough to receive NIVO + IPI following the failure of anti-PD-1 monotherapy, it might be more relevant to prescribe the full-dose IPI regimen.

Looking at the different types of irAE encountered, the toxicity profile is in agreement with the previous reports [28], except for the case of myocarditis. Indeed, the rate of immune-related myocarditis in the group of patients treated with NIVO1 + IPI3 was high (2/27, 7%), being much higher than the rate of 2.4% previously reported in the literature [29]. The interpretation of this finding is limited by the small size of this subgroup but this irAE may represent an issue in this population because of the high prevalence of cardiovascular comorbidities such as diabetes and hypertension which are suspected risk factors for immune-related myocarditis [30]. Moreover, cardiomyocyte senescence may also favor the onset of this AE in the elderly population. As observed herein in the two patients, the onset of myocarditis after immunotherapy initiation is usually rapid [31] and therefore close cardiac monitoring is recommended, especially during the first 6 weeks of treatment.

This study is limited by its retrospective nature, the small size of the cohort, the heterogeneity in prescribing patterns and the absence of a comparison with younger patients treated in the exact same setting. The safety data reported in this cohort are thus very preliminary and should be interpreted with caution. However, the safety profile of this cohort is in accordance with the available data related to the use of NIVO + IPI in elderly patients, showing no evidence of poorer tolerance in older age groups, regardless of the tumor type [32]. In addition, a recent retrospective study demonstrated no difference in terms of efficacy and toxicity for patients aged over 75 years in comparison to those for younger counterparts [33].

Regarding the line of treatment, several trials conducted in younger melanoma patients have demonstrated the significant antitumor activity of anti-PD-1 + ipilimumab (either in an amount of 1 mg/kg [34] or 3 mg/kg [35]) after anti-PD-1/L1 immunotherapy failure. However, these encouraging results may not be applicable to the elderly population since a recent study also pointed out that the survival benefit for patients aged over 75 years treated with checkpoint inhibitor monotherapy was mainly observed in the first-line setting [36]. In our study, no conclusion can be drawn about the therapeutic value of NIVO + IPI as a salvage therapy since we did not assess the efficacy of this combination. Given the substantial risk of toxicity and the cost of NIVO + IPI, this point requires future investigation.

## 5. Conclusions

The present study raises the question of an age limit for the use of NIVO + IPI. Part of the answer depends on ethical and economic considerations, and there is thus no consensual recommendation on that point. Nevertheless, when considering this combination in very elderly patients, it does seem reasonable to favor its use for patients with an ECOG PS of 0–1. The results of our study show a safety profile of the combination that seems acceptable in the very elderly population; whether or not the low-dose IPI regimen NIVO3 + IPI1 should be preferred over NIVO1 + IPI3 remains an open question. Further investigations with prospective trials specifically designed for very elderly patients are warranted for the better management of this growing yet understudied population.

## Figures and Tables

**Table 1 cancers-15-04330-t001:** Baseline characteristics.

Baseline Characteristics	Regimen Group, No. (%)			
	NIVO + IPI N = 60	NIVO1 + IPI3 N = 27	NIVO3 + IPI1 N = 33	*p* Value *
**Age**				
Age, mean (SD)—years	83.7 (3.2)	82.4 (1.7)	84.7 (3.7)	0.002
Age, median [range]—years	82.9 [79.3–93.3]	82.1 [79.7–85.4]	83.4 [79.3–93.3]	
**Sex**				0.5
Male	37 (62)	15 (56)	22 (67)	
Female	23 (38)	12 (44)	11 (33)	
**ECOG prior to treatment**				0.1
ECOG score 0–1	47 (78)	22 (81)	25 (76)	
ECOG score 2–3	13 (22)	5 (19)	8 (24)	
**Independent home-living**				0.4
No	5 (8)	1 (4)	4 (12)	
Yes	55 (92)	26 (96)	29 (88)	
**Comorbidities**				0.3
No	4 (7)	3 (11)	1 (3)	
Yes	56 (93)	24 (89)	32 (97)	
Hypertension	40 (67)	19 (70)	21 (64)	
Diabetes	9 (15)	3 (11)	6 (18)	
Overweight	22 (37)	9 (33)	13 (39)	
Dementia	3 (5)	1 (4)	2 (6)	
CID	2 (3)	0	2 (6)	
**Polymedication ****				0.8
No	40 (67)	17 (63)	23 (70)	
Yes	20 (33)	10 (37)	10 (30)	
**Melanoma subtype**				0.6
Cutaneous	37 (62)	15 (55)	22 (67)	
Mucosal	12 (20)	7 (26)	5 (15)	
Acral	3 (5)	1 (4)	2 (6)	
Uveal	1 (2)	0	1 (3)	
Unknown primary	7 (11)	4 (15)	3 (9)	

Abbreviations: NIVO1 + IPI3, nivolumab 1 mg/kg + ipilimumab 3 mg/kg; NIVO3 + IPI1, nivolumab 3 mg/kg + ipilimumab 1 mg/kg; y, years; ECOG, Eastern Cooperative Oncology Group; CID, chronic inflammatory disease; SSM, superficial spreading melanoma; AJCC, American Joint Commission on Cancer; LDH, lactate dehydrogenase; ULN, upper limit of normal. * Comparison between the NIVO1 + IPI3 group and NIVO3 + IPI1 group. ** Polymedication defined as a daily intake of five or more medications.

**Table 2 cancers-15-04330-t002:** Global safety profile.

	Regimen Group, No. (%)			
	NIVO + IPI N = 60	NIVO1 + IPI3 N = 27	NIVO3 + IPI1 N = 33	*p* Value
Any treatment-related AE	38 (63)	18 (67)	20 (61)	0.83
Mean time of onset (SD)—days	36 (28.9)	23 (18.4)	49 (31.6)	0.003
AE Grade ≥ 3	16 (27)	12 (44)	4 (12)	0.04
Discontinuation for toxicity	17 (28)	13 (48)	4 (12)	0.003

Abbreviations: NIVO1 + IPI3, nivolumab 1 mg/kg + ipilimumab 3 mg/kg; NIVO3 + IPI1, nivolumab 3 mg/kg + ipilimumab 1 mg/kg; AE, adverse event; SD, standard deviation.

**Table 3 cancers-15-04330-t003:** Treatment-related AEs that occurred in at least 10% of patients.

	Regimen Group, No. (%)	
	NIVO1 + IPI3 N = 27	NIVO3 + IPI1 N = 33
Pruritus and/or Rash	6 (22)	7 (21)
Grade 3–4	1 (4)	0 (0)
Colitis	7 (26)	5 (15)
Grade 3–4	4 (15)	1 (3)
Endocrinopathy *	5 (19)	6 (18)
Grade 3–4	2 (7)	3 (9)
Hepatitis	6 (22)	2 (6)
Grade 3–4	4 (15)	0 (0)
Arthritis/arthralgia	3 (11)	2 (6)
Grade 3–4	0 (0)	0 (0)
Death **	2 (7)	0 (0)

Abbreviations: NIVO1 + IPI3, nivolumab 1 mg/kg + ipilimumab 3 mg/kg; NIVO3 + IPI1, nivolumab 3 mg/kg + ipilimumab 1 mg/kg; AE, adverse event. * Endocrinopathy includes thyroiditis, adrenal insufficiency and hypophysitis. ** Two fulminant myocarditis cases.

**Table 4 cancers-15-04330-t004:** Safety profile in the first-line setting.

	First-Line Setting, Regimen Group, No. (%)			
	NIVO + IPI N = 39	NIVO1 + IPI3 N = 25	NIVO3 + IPI1 N = 14	*p* Value
Any treatment-related AE	28 (72)	18 (72)	10 (71)	1
AE Grade ≥ 3	13 (33)	12 (48)	1 (7)	0.013

Abbreviations: NIVO1 + IPI3, nivolumab 1 mg/kg + ipilimumab 3 mg/kg; NIVO3 + IPI1, nivolumab 3 mg/kg + ipilimumab 1 mg/kg; AE, adverse event.

**Table 5 cancers-15-04330-t005:** Patients’ outcomes at first evaluation in the first-line setting.

	First-Line Setting, Regimen Group, No. (%)		
	NIVO + IPI N = 39	NIVO1 + IPI3 N = 25	NIVO3 + IPI1 N = 14
Complete response	0	0	0
Partial response	15 (38.5)	10 (40)	5 (36)
Stable disease	3 (8)	2 (8)	1 (7)
Progression	15 (38.5)	8 (32)	7 (50)
Death	6 (15)	5 (20)	1 (7)

Abbreviations: NIVO1 + IPI3, nivolumab 1 mg/kg + ipilimumab 3 mg/kg; NIVO3 + IPI1, nivolumab 3 mg/kg + ipilimumab 1 mg/kg.

## Data Availability

The anonymised datasets of this study in xlsx format are available from the corresponding author upon reasonable request. The data are not publicly available due to privacy.

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
