# Peer review of "Combined Nivolumab and Ipilimumab in Octogenarian and Nonagenarian Melanoma Patients"

_cancers, 2023, doi:10.3390/cancers15174330_

Round 1

Reviewer 1 Report

Thank you for allowing me to review. The article covers an important aspect of melanoma management and such data is very relevant. The methodology is sound, as are the results. I only have two minor corrections:

1- Figure 1 is redundant and can be removed (the data is covered in table 1)

2- Can the authors please present the efficacy data in a table form, including response percentages. Although very underpowered, a statistical comparison may also be useful. Furthermore, if resources permit, any survival data would also be very useful.

Reviewer 2 Report

The authors investigated the safety of two dosage regimens of nivolumab combined with ipilimumab in older adults. The topic is of particular interest due to the limited data available. However, if it proves challenging to provide additional data, its practical application in a clinical setting might be limited. I also have concerns regarding the authors' interpretations of their findings.

While the authors compare two different dosing regimens, this data comparison in itself doesn't establish the safety of the combination therapy in the elderly. Shouldn't there be a comparison of safety between older and younger patients for a more comprehensive understanding? While differences in treatment lines between the two groups might complicate a comparison that includes efficacy, shouldn't there be more discussion on which regimen is better suited for older patients?

Despite the limited patient cohort, the fact that 2 out of 27 patients in the Nivo1+Ipi3 group experienced deaths due to irAEs seems to contradict the conclusion of its safety in the elderly population.

Reviewer 3 Report

1. Please expand in the introduction section about why it is important to delineate safety patterns in octogenarian and nonagenarian melanoma patients. What variables could specifically contribute to examining their safety profile compared to older adults (>65years).

2. Is heterogeneity in prescription patterns exclusive to octogenarian and nonagenarian melanoma patients? How do the multiple prescription patterns compare to other older adults.

3. The authors tout safety as the main conclusion, however the safety profile of this retrospective study population is confounded by previous treatment with anti-PD1 and poly-medication. More could be discussed about safety in the 65% of patient population that was not exposed to previous lines of treatment as a more accurate assessment of safety.

4. The study seems to be under-powered to make any conclusions about efficacy while conclusions about safety have multiple confounding factors.

Round 2

Reviewer 3 Report

Nice updates.